# Better Together: Resnet-50 accuracy with $13\times$ fewer parameters and at $3\times$ speed

## Abstract

Recent research on compressing deep neural networks has focused on reducing the number of parameters. Smaller networks are easier to export and deploy on edge-devices. We introduce *Adjoined networks* as a training approach that can regularize and compress any CNN-based neural architecture. Our one-shot learning paradigm trains both the original and the smaller networks together. The parameters of the smaller network are shared across both the architectures. We prove strong theoretical guarantees on the regularization behavior of the adjoint training paradigm. We complement our theoretical analysis by an extensive empirical evaluation of both the compression and regularization behavior of adjoint networks. For resnet-50 trained adjointly on Imagenet, we are able to achieve a $13.7x$ reduction in the number of parameters[1] and a $3x$ improvement in inference time without any significant drop in accuracy. For the same architecture on CIFAR-100, we are able to achieve a $99.7x$ reduction in the number of parameters and a $5x$ improvement in inference time. On both these datasets, the original network trained in the adjoint fashion gains about $3\%$ in top-1 accuracy as compared to the same network trained in the standard fashion.

## 1 Introduction

Neural networks have achieved state-of-the art performance on computer vision such as classification, object detection Redmon et al. (2016), image segmentation Badrinarayanan et al. (2017) and many more. Since the introduction of Alexnet Krizhevsky et al. (2012), neural architectures have progressively gone deeper with an increase in the number of parameters. This includes architectures like Resnet He et al. (2016) and its many variants (xresnet He et al. (2019), resnext Xie et al. (2017); Hu et al. (2018) etc.), Densenet Huang et al. (2017), Inception networks Chen et al. (2017) and many others.

While these networks achieve exceptional performance on many tasks, their large size makes it difficult to deploy on many edge devices (like mobile phones, iot and embedded devices). Unlike cloud servers, these edge devices are constrained in terms of memory, compute and energy resources. A large network performs a lot of computations, consumes more energy and is difficult to transport and to update. A large network also has a high prediction time per image. This is constraint when real-time inference is needed. Thus, compressing neural networks while maintaining accuracy has received significant attention in the last few years.

**Pruning** - These techniques involve removing parameters (or weights) which satisfy some criteria. For example, in weight pruning, all the parameters whose values are below some pre-determined threshold are removed Han et al. (2015). A natural extension of this is channel pruning Liu et al. (2017) and filter pruning Li et al. (2016) where entire convolution channel or filter is removed according to some criteria. However, all of these methods involve multiple passes of pruning followed by fine-tuning and require a very long time to compress. Moreover, weight pruning doesn't give the benefit of faster inference times unless there is hardware support for fast sparse matrix multiplications. In this paper, we propose a one-shot compression procedure (as opposed to multiple passes). To the best of our knowledge, pruning techniques have not been successfully applied (without significant accuracy loss) to large architectures like Resnet-50 considered in this paper.

---

[1]For size comparison, we ignore the parameters in the last linear layer as it varies by dataset and are typically dropped during fine-tuning. Else, the reductions are $11.5x$ and $95x$ for imagenet and cifar-100 respectively.

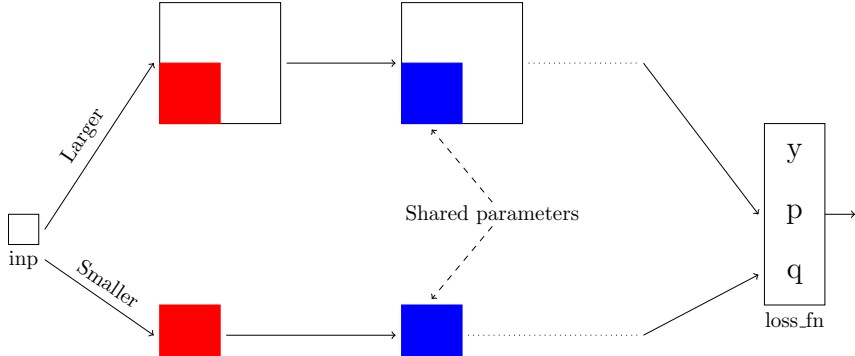

Figure 1: Training paradigm based on adjoined networks. The original and the compressed version of the network are trained together with the parameters of the smaller network shared across both. The network outputs two probability vectors $p$ (original network)and $q$ (corresponding to the smaller network).

**Quantization and low-precision training** - In quantization-based techniques, multiple parameters are quantized such that share the same value. Hence, only the effective values and their indices need to be stored Han et al. (2015). This method of scalar quantization can be extended to vector quantization where a group of parameters share values Carreira-Perpinán & Idelbayev (2017). In a recent work, Stock et al. (2019) used a clustering based approach for vector quantization of resnet architectures. A similar approach is low-precision training where the goal is to train networks with integer or ternary or binary weights instead of floating point numbers Shayer et al. (2017); McDonnell (2018); Courbariaux et al. (2015). As before, these techniques assume the availability of special hardware which supports fast inference.

**Small architectures** - Another approach is to design smaller architectures which can run efficiently on edge devices. SqueezeNet Iandola et al. (2016), MobileNet Sandler et al. (2018) and EfficientNet Tan & Le (2019) to name a few. In this paper, our goal is to design a paradigm which can compress any architecture. Hence, the direction of architecture search is orthogonal to our approach.

To summarize, most of the current approaches suffer from one of the two problems. (1) Require the availability of special hardware to support fast inference. (2) Require large training time as they alternate between pruning and fine-tuning. In this work, we propose a novel training paradigm based on *adjoined networks* which can compress any neural architecture, provides inference-time speedups and works at the application layer (does not require any specialized hardware).

As shown in Fig. 1, in the adjoint training paradigm, both the original and the compressed network are trained together at the same time. The parameters of the larger network are a super-set of the parameters of the smaller network. Details of our design, how it supports fast inference and relationship with other architectures (teacher-student Hinton et al. (2015), siamese networks Bertinetto et al. (2016), slimmable networks Yu et al. (2018), deep mutual learning Zhang et al. (2018) and lottery ticket hypothesis Frankle & Carbin (2018)) are discussed in Section 2. In our training paradigm, we get two outputs $p$ and $q$ corresponding to the original and smaller networks respectively. We train the two networks using a novel time-dependent loss function, *adjoint loss* described in Section 3. The adjoint loss not only trains the smaller network but also acts as a regularizer for the bigger (original) network. In Section 4, we provide strong **theoretical guarentees** on the regularization behaviour of adjoint training. We also show that training and regularizing in the adjoint fashion is better than other regularization techniques like dropouts Srivastava et al. (2014).

In Section 5, we describe our results. We run several experiments on various datasets like ImageNet Russakovsky et al. (2015) and CIFAR-10 and CIFAR-100 Krizhevsky et al. (2009). For each of these datasets, we consider different architectures like resnet-50 and resnet-18. On CIFAR-100, the adjoint training paradigm allows to compress resnet-50 architecture by $99.7x$ without losing any accuracy. The compressed architecture has an inference speed of $5x$ when compared against the original, bigger architecture. Moreover, the original network gains $3.58\%$ in accuracy when compared against the same network trained in the standard (non-adjoint) fashion. On the same dataset, for resnet-18, we

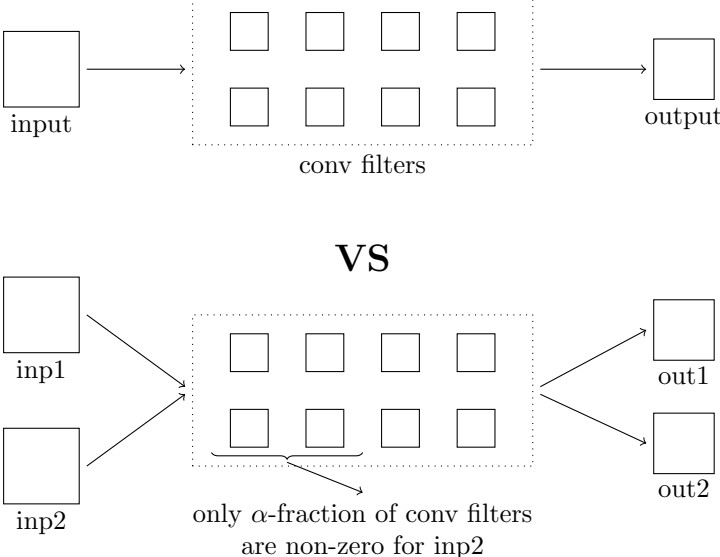

only $\alpha$-fraction of conv filters
are non-zero for inp2

Figure 2: (Top) Standard $2d$-convolution operation. The inputs, outputs and convolution filters are all $3d$ volumes. (Bottom) Adjoint convolution operation. The convolution layer receives two inputs inp1 and inp2. Standard convolution operation is applied on inp1 to get out1. For the second input, only a fraction of the conv filters are used (rest can be treated as zero or omitted) to get out2.

acheive a compression factor of $49x$ at an inference speed of $3x$ with the bigger network gaining $5.55\%$ accuracy (over the non-adjoint counterpart). On Imagenet, for resnet-50, we are able to compress it by $13.7x$ at an inference speed of $3x$. In this case, the bigger network gains $2.43\%$ over its non-adjoint cousin clearly showing that it is **better** to train the networks **together**.

## 2 ADJOINED NETWORKS

In our training paradigm the original (bigger) and the smaller network are trained together. The motivation for this kind of training comes from the principle that *good teachers are lifelong learners*. Hence, the bigger network which serves as a teacher for the smaller network should not be frozen (as in standard teacher-student architecture designs Hinton et al. (2015)). But rather both should learn together in a 'combined learning environment', that is, adjoined networks.[2] By learning together both the networks can be better together.

We are now ready to describe our approach and discuss the design of adjoined networks. But before that, let take a re-look at the standard convolution operator. Let $\mathbf{x} \in \mathbf{R}^{h \times w \times c_{in}}$ be the input to a convolution layer with weights $\mathbf{W} \in \mathbf{R}^{c_{out} \times k \times k \times c_{in}}$ where $c_{in}, c_{out}$ denotes the number of input and output channels, $k$ the kernel size and $h, w$ the height and width of the image. Then, we have that

$$\mathbf{y} = conv(\mathbf{x}, \mathbf{W})$$

In the adjoint paradigm, a convolution layer with weight matrix $\mathbf{W}$ and a binary mask matrix $M \in \{0,1\}^{c_{out} \times k \times k \times c_{in}}$ receives two inputs $\mathbf{x}_1$ and $\mathbf{x}_2$ of size $h \times w \times c_{in}$ and outputs two vectors $\mathbf{y}_1$ and $\mathbf{y}_2$ as defined below.

$$\mathbf{y}_1 = conv(\mathbf{x}_1, \mathbf{W}) \qquad\qquad \mathbf{y}_2 = conv(\mathbf{x}_2, \mathbf{W} * M) \qquad (1)$$

Here $M$ is of the same shape as $\mathbf{W}$; and $*$ represents an element-wise multiplication. note that the parameters of the matrix $M$ are fixed before training and not learnt. The vector $\mathbf{x}_1$ represents an input to the original (bigger) network while the vector $\mathbf{x}_2$ is the input to the smaller, compressed network. For the first convolution layer of the network $\mathbf{x}_1 = \mathbf{x}_2$. But the two vectors are not necessarily equal for the deeper convolution layers (Fig. 1). The mask matrix $M$ serves to zero-out some of the parameters of the convolution layer thereby enabling network compression. In this paper, we consider matrices $M$ of the following form.

$$M := a_\alpha = \text{ matrix such that the first } \frac{c_{out}}{\alpha} \text{ rows are all 1 and the rest 0} \qquad (2)$$

---

[2]Throughout the paper, we use the term adjoined networks and adjoint training paradigm interchangeably.

Consider another matrix $r_\beta$ of the same shape as $M$ which is such that $\beta$ fraction of the entries (selected uniformly at random) are zero and the rest are one. We achieve further compression, by multiplying $a_\alpha$ by $r_\beta$. In Section 5, we run experiments with $M := a_\alpha$ for $\alpha \in \{4, 8, 16\}$ and for various choices $\beta \in \{0.1, \ldots, 0.9\}$. Putting this all together, we see that any CNN-based architecture can be converted and trained in an adjoint fashion by replacing the standard convolution operation by the adjoint convolution operation (Eqn. 1). Since the first layer receives a single input (Fig. 1), two copies are created which are passed to the adjoined network. The network finally gives two outputs $\mathbf{p}$ corresponding to the original (bigger or unmasked) network and $\mathbf{q}$ corresponding to the smaller (compressed) network where each convolution operation is done using a subset of the parameters described by the mask matrix $M$ (or $M_\alpha$). We train the network using a novel time-dependent loss function which forces $\mathbf{p}$ and $\mathbf{q}$ to be close to one another (Defn. 1). Before we describe our loss function in detail, we first compare our approach to other similar approaches and ideas in the literature.

**Similar architectures** - The idea of training two networks together was recently considered by Zhang et al. (2018). Rather than using a teacher to train a student, they let a cohort of students train together using a distillation loss function. In this paper, we consider we train a teacher and a student together rather than a cohort of students. We also use a novel time-dependent loss function. Moreover, we also provide theoretical guarantees on the efficacy of our approach. Another related architecture was considered in the work of Slimmable Networks Yu et al. (2018). Here different networks (or architectures) are *switched on* one-at-a-time and trained using the standard logistic loss function. By contrast, in this work, both the networks are trained together at the same time using a novel loss function (adjoint-loss). Moreover, in this work, we show compression and regularization results on large-scale datasets like Imagenet and for large architectures like Resnet-50.

**Siamese networks** is a neural network design that is used to compare two input vectors. Also called twin networks, they consist of two networks with identical parameters. Given two input vectors, the network returns two output vectors which are then used to compute the similarity between the input vectors Bromley et al. (1994). More recently, siamese networks have been used in face verification Taigman et al. (2014). The design of adjoined network also uses two architectures. However, both the architectures are not identical. Rather one is a super-set of the other. Also, rather than working on two different input vectors, adjoined networks work on a single input.

**Lottery ticket hypothesis** Frankle & Carbin (2018) says the following. A neural network contains within it a subset of parameters which are 'special'. These $10 - 20\%$ special parameters won the lottery during the network initialization process and it is possible to prune and train the original network to using only these special parameters. Thus, some weights in the network are just 'lucky' that they happened to be initialized in the 'right' way which made training possible with only these subsets of parameters. In our design, rather than relying on random initialization, we "force" some parameters to be special through the use of the mask matrix $M$ and then training the two networks together through the adjoint-loss (Defn. 1).

## 3 ONE-SHOT REGULARIZATION AND COMPRESSION

In the previous section, we looked at the design on adjoined networks. For one input $(\mathbf{X}, \mathbf{y}) \in \mathbf{R}^{h \times w \times c_{in}} \times [0, 1]^{n_c}$, the network outputs two vectors $\mathbf{p}$ and $\mathbf{q} \in [0, 1]^{n_c}$ where $n_c$ denotes the number of classes and $c_{in}$ denotes the number of input channels (equals $3$ for RGB images).

**Definition 1** (Adjoint loss). *Let $y$ be the ground-truth one-hot encoded vector and $p$ and $q$ be output probabilities by the adjoined network. Then*

$$\mathcal{L}(y, p, q) = -y \log p + \lambda(t) \, KL(p, q) \tag{3}$$

*where $KL(p, q) = \sum_i p_i \log \frac{p_i}{q_i}$ is the measure of difference between two probability measures Kullback & Leibler (1951). The regularization term $\lambda : [0, 1] \to \mathbf{R}$ is a function which changes with the number of epochs during training. Here $t = \frac{current\ epoch}{Total\ number\ of\ epochs}$ equals zero at the start of training and equals one at the end.*

In our definition of loss function, the first term is the standard cross-entropy loss function which trains the bigger network. To train the smaller network, we use the predictions from the bigger network as a soft ground-truth signal. We use kl-divergence to measure how far the output of the smaller network

is from the bigger network. This also has a regularizing effect as it forces the network to learn from a smaller set of parameters. Note that, in our implementations, we use $KL(p, q) = \sum p_i \log \frac{p_i + \epsilon}{q_i + \epsilon}$ to avoid rounding and division by zero errors where $\epsilon = 10^{-6}$.

At the start of training, $p$ is not a reliable indicator of the ground-truth labels. To compensate for this, the regularization term $\lambda$ changes with time. In our experiments, we used $\lambda(t) = \min\{4t^2, 1\}$. Thus, the contribution of the second term in the loss is zero at the beginning and steadily grows to one at $50\%$ training. We experiment with different choices of the regularization function $\lambda$ the results of which are in the appendix .

**Comparison with dropout** - Our idea of using only a subset of the parameters has some similarity to using dropouts where a fraction of the parameters are initialized to zero at training time. In this case, only the output of the smaller network is used to train the parameters. In our case, we use both the outputs. Our experiments show that training the networks together is a much more effective regularization strategy as compared to dropouts.

## 4 THEORY: REGULARIZATION BEHAVIOR

To study the regularization behavior of adjoint training paradigm (or the loss function), we consider the following neural network model. A multi-layer neural network that has only convolution or linear fully-connected layers with two possible choices for activation function, ReLU or the linear function. Although neural networks used in practice often have other types of layers max-pooling, batch-normalization etc. and other possible activation functions, the proposed model is often studied as a first step towards understanding the dynamics and behavior of deep networks (for example in Phuong & Lampert (2019), Saxe et al. (2013), Kawaguchi (2016) and Hardt & Ma (2016)). We are now ready to state the main theoretical result of this paper.

**Theorem 4.1.** *Given a deep neural network $\mathcal{A}$ which consists of only convolution and linear layers. Let the network use one of $f(x) = \min\{x, 0\}$ (relu) or $f(x) = x$ (linear) as the activation function. Let the network be trained using the adjoint loss function as defined in Eqn. 3. Let $\mathbf{X}$ be the set of parameters of the network $\mathcal{A}$ which is shared across both the smaller and bigger networks. Let $\mathbf{Y}$ be the set of parameters of the bigger network not shared with the smaller network. Let $\mathbf{p}$ be the output of the larger network and let $\mathbf{q}$ be the output of the smaller network. Then, the adjoint loss function induces a data-dependent regularizer with the following properties.*

- *For all $x \in X$, the induced $L_2$ penalty is given by $\sum_i \mathbf{p}_i \left( \log' \mathbf{p}_i - \log' \mathbf{q}_i \right)^2$*
- *For all $y \in Y$, the induced $L_2$ penalty is given by $\sum_i \mathbf{p}_i \left( \log' \mathbf{p}_i \right)^2$*

**Proof sketch** To analyze the induced $L_2$ penalty, we consider the second order taylor approximation of the kl-divergence term for all parameters $x \in \mathbf{X}$ and for all $y \in \mathbf{Y}$. By treating, $p_i$ (and $q_i$) as functions of the given parameter $x$ (or $y$) and careful analysis gives us the result of the theorem. The detailed derivations are in the appendix . □

Thm. A.1 shows that for that shared parameters $x$, adjoint loss penalizes those parameters whose rate of change between the larger and smaller networks is large (on the log scale). Hence, it encourages' parameters that have similar behavior (rate of change) on both the networks. Similarly, for the over-parameterized weights of the model $y$, the regularization imposed by the adjoint loss is such that if these parameters change a lot (on the log scale) then the penalty imposed on such parameters is more. Thus, adjoint loss "encourages" such parameters not to change by a lot thereby enabling compression.

## 5 EXPERIMENTS

We are now ready to describe our experiments in detail. We run experiments on five different datasets. (1) *Imagenet* - an image classification dataset Russakovsky et al. (2015) with 1000 classes and about $1.2M$ images . (2) *CIFAR-10* - a collection of $60k$ images in 10 classes. (3) *CIFAR-100* - same as cifar-10 Krizhevsky et al. (2009) but with 100 classes. (4) *Imagewoof* - A proxy dataset Howard (2019); Shleifer & Prokop (2019) containing 10 different dog breeds from imagenet. (5) *Oxford-IIIT Pets* - a dataset Parkhi et al. (2012) of pets containing 37 classes with approximately 200 images per

| Adjoint-small vs teacher - student vs standard | | | | | | | |
|---|---|---|---|---|---|---|---|
| Network | Dataset | Mask matrix ($M$) | #params | Speed gain | Adjoint (small) | Teacher student | Standard |
| Resnet-50 | Cifar-10 | $a_8 * r_{0.9}$ | $88.9\times$ | $4\times$ | 90.05 | 89.44 | 90.25 |
| | Cifar-100 | $a_{16} * r_{0.9}$ | $99.6\times$ | $4.7\times$ | 64.54 | 63.62 | 65.31 |
| | Imagenet | $a_4$ | $13.7\times$ | $3.1\times$ | 71.84 | 70.55 | $73.41^4$ |
| | Imagewoof | $a_4 * r_{0.8}$ | $44.8\times$ | $3.1\times$ | 84.76 | 83.88 | 85.2 |
| | Pets | $a_4 * r_{0.5}$ | $24.15\times$ | $3.1\times$ | **85.38** | 84.97 | 85.31 |
| Resnet-18 | Cifar-10 | $a_{16} * r_{0.9}$ | $68.7\times$ | $2.6\times$ | 87.67 | 88.43 | 88.17 |
| | Cifar-100 | $a_4 * r_{0.9}$ | $48.9\times$ | $2.1\times$ | **61.42** | 60.69 | 61.39 |
| | Imagewoof | $a_4 * r_{0.5}$ | $21.8\times$ | $2.1\times$ | 82.81 | 82.61 | 83.35 |
| | Pets | $a_2 * r_{0.5}$ | $7.2\times$ | $1.7\times$ | 83.62 | 80.98 | 84.84 |

Table 1: The #params column is the ratio of number of parameters of the smaller network compared against the standard full network. The speed gain column denotes the ratio of inference time. The last three columns compare the accuracies (in %) of the the smaller network trained using the adjoint paradigm vs the the accuracy of the same network trained using the teacher student paradigm against the accuracy of the standard full network. $a_\alpha$ is as defined in Eqn. 2 and $r_\beta$ denotes a random matrix with $\beta$ fraction of its entries are zero. $*$ denotes element-wise dot-product. Detailed results can be found in the appendix .

class. For each of these datasets, we use standard data augmentation techniques like random-resize cropping, random flipping etc. The details are provided in the appendix .

We train two different architectures on all of the above datasets. Namely, resnet50 and resnet18. The detailed architecture diagram can be found in supplementary material. On each dataset, we first train these architectures in the standard non-adjoint fashion using the cross-entropy loss function. We will refer to it by the name *standard* or *standard-full*. Next, we train the adjoint network, obtained by replacing the standard convolution operation by the adjoint convolution operation, using the adjoint loss function. In the second step, we obtain two different networks. In this section, we refer to them by *ajoint-full* and the *adjoint-small* networks. We compare the performance of the adjoint-full and adjoint-small networks against the standard network. One point to note is that we do not replace the convolutions in the stem layers but only those in the res-blocks. Since most of the weights are in the later layers, this leads to significant space and time savings while retaining competitive accuracy.

We ran our experiments on gpu enabled machine using pytorch. We trained all our networks using the adam optimizer with a cosine learning rate schedule with gradual warm-up. The parameters of the network were randomly initialized. Unless otherwise specified, we train both the standard and adjoint networks for the same number of epochs. We have also open-sourced our implementation [3].

In Section 5.1, we describe our results for compression. In Section 5.2, we show the strong regularizing effect of adjoint training. In Section 5.3, we compare our approach to dropouts, a popular approach to regularizing deep neural networks. In total, we ran close to 150 different experiments covering different datasets, different masking matrices etc. The detailed results are included in the appendix .

## 5.1 COMPRESSION

Table 1 compares the performance (top 1% accuracy)of the adjoint-small network against the performance of standard-full network. We use the $a_\alpha$ as the masking matrix (defined in Eqn. 2). The mask is such that the last $(1 - \frac{1}{\alpha})$ filters are zero. Hence, these can be pruned away to support fast inference. Using $a_\alpha$ suffices for speed optimization. But, we can get a further reduction in the number of parameters, by multiplying the adjoint matrix by another random matrix $r$. This matrix is such that only a fraction of its entries are non-zero. Hence, using such a mask matrix further reduces the model size.

---

[3]The code can be found at `https://github.com/utkarshnath/Adjoint-Network.git`
[4]We train on imagenet using the standard pytorch code Facebook (2020). The accuracy is the same as reported for multi-gpu training on imagnet by nvidia Nvidia (2020)

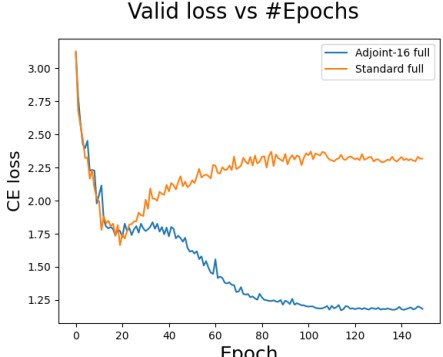 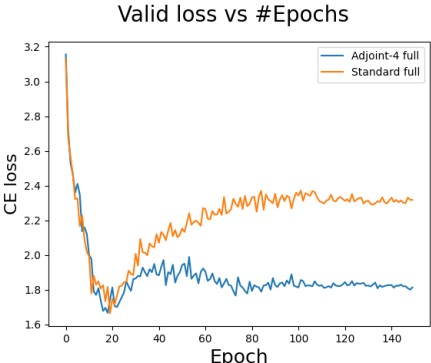

Figure 3: (Left) Plot of validation cross-entropy loss of the adjoint-16 full and standard resnet50 network on CIFAR-100. (Right) Plot of validation cross-entropy loss of the adjoint-4 full and standard resnet50 network on Imagenet

| Adjoint-full vs standard | | | | |
|---|---|---|---|---|
| Network | Dataset | Mask matrix $(M)$ | Adjoint-full | Standard |
| Resnet-50 | Cifar-10 | $a_{16}$ | 91.34 | 90.25 |
| | Cifar-100 | $a_{16}$ | **69.07** | 65.31 |
| | Imagenet | $a_4$ | **75.84** | 73.41 |
| | Imagewoof | $a_4 * r_{0.9}$ | 86.3 | 85.2 |
| | Pets | $a_2$ | 87 | 85.31 |
| Resnet-18 | Cifar-10 | $a_8$ | 90.26 | 88.17 |
| | Cifar-100 | $a_4$ | **66.84** | 61.39 |
| | Imagewoof | $a_4$ | 84.16 | 83.35 |
| | Pets | $a_8$ | **86.33** | 84.84 |

Table 2: The last two columns show the accuracies (in %) of the network trained in the adjoint fashion and the same network trained in the standard way. In all cases the adjoint network exceeds the accuracy of the standard full network. $a_\alpha$, $r_\beta$ are as in Table 1. Detailed results can be found in the appendix .

We also observe that resnet-50 is a bigger network and can be compressed more. Also, different datasets can be compressed by different amounts. For example, on cifar-10 and 100 datasets, the network can be compressed by factors $\sim 90x$ while for other datasets it ranges from $7x$ to $44x$. Our goal is to compare the adjoint models against the standard models. Note that in all the cases, the drop in accuracy is small, a maximum of $-1.4\%$ over all the datasets. In some cases, the smaller network even outperforms the bigger network. We also compare the performance of the smaller network trained in the adjoint fashion against the same network trained using the standard teacher-student paradigm. On all the datasets and for all architectures (except one) that we considered, adjoint training paradigm out-performed teacher student training.

## 5.2 REGULARIZATION

Table 2 compares the performance of the adjoint-full network against the performance of the corresponding standard-full network. We see a consistent trend that the network trained adjointly outperforms the same network trained in the standard way. We see maximum gains on cifar-100, exceeding accuracies by as much as $5.5\%$. Even on imagenet, we see a gain of about $2.5\%$. Fig. 3 shows the plot of validation cross-entropy loss as a function of the number of epochs. For a fair comparison, we look at the $-y \log p$ (Eqn. 3) term of the adjoint loss function. By regularizing the loss function, we are able to train longer while decreasing the loss function. On the other hand, the network trained in the standard fashion starts to over-fit after a while (as evident in the loss profile). Fig. 3 shows the loss plot on one architecture for two datasets. Similar plots on other datasets and for other architectures are available in the appendix .

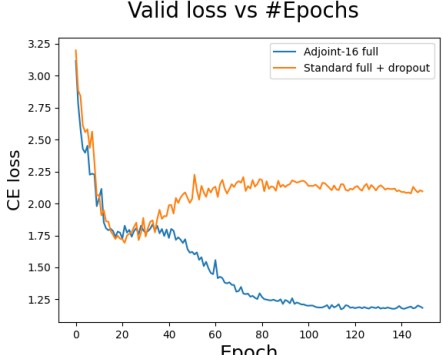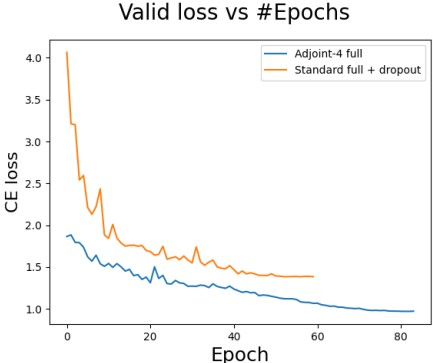

Figure 4: (Left) Plot of validation cross-entropy loss of the adjoint-16 full and standard + dropout resnet50 network on CIFAR-100. (Right) Plot of validation cross-entropy loss of the adjoint-4 full and standard + dropout resnet50 network on Imagenet

| Adjoint-full vs standard + dropouts | | | | | |
|---|---|---|---|---|---|
| Network | Dataset | Mask matrix ($M$) | | Full | Std + dropout |
| Resnet-50 | Cifar-10 | $a_{16}$ | | 91.34 | 90.11 |
| | Cifar-100 | $a_{16}$ | | **69.07** | 66.66 |
| | Imagenet | $a_4$ | | **75.84** | 69.54 |
| | Imagewoof | $a_4 * r_{0.9}$ | | **86.3** | 82.76 |
| | Pets | $a_4$ | | 86.73 | 84.7 |
| Resnet-18 | Cifar-10 | $a_8$ | | 90.26 | 89.04 |
| | Cifar-100 | $a_4$ | | 66.09 | 64.3 |
| | Imagewoof | $a_4$ | | 84.16 | 82.1 |
| | Pets | $a_8$ | | **86.33** | 82.61 |

Table 3: The last two columns show the accuracies (in %) of the network trained in the adjoint fashion vs the same network trained in the standard way using dropouts. In all cases the adjoint network exceeds the accuracy of the standard one. $a_\alpha$, $r_\beta$ are as in Table 1. Details can be found in the appendix .

### 5.3 COMPARISON AGAINST DROPOUTS

Table 3 compares the performance of the adjoint-full network against the performance of the corresponding standard-full network where the standard network was trained using dropouts. We see a consistent trend that the network trained adjointly outperforms the same network trained with dropout regularization. We see maximum gains on imagenet, exceeding accuracies by as much as $6.3\%$. On cifar and imagewoof, we see gains of about $3\%$. As is evident from our experiments, dropouts are not very effective regularizers on the imagenet dataset. Fig. 4 shows the plot of validation cross-entropy loss as a function of the number of epochs. As before, we only look at the $-y \log p$ (Eqn. 3) term. The above results show that adjoint training is much more effective regularization strategy as compared against dropouts. Similar plots on other datasets and for other architectures are available in the appendix .

## 6 CONCLUSION

In this work, we introduced the paradigm of Adjoined network training. We showed how this approach to training neural networks can allow us to compress large networks like Resnet-50 by $13x$, (even going up to $99.7x$ on some datasets) while retaining the accuracy. We showed that the idea of adjoining two networks together can be used to regularize any architecture. We showed that adjoining a large and a small network together enables the large network to significantly exceed its own accuracy (when trained in the standard way) and is a much more effective regularization strategy than dropouts.

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

## A  REGULARIZATION THEORY

**Theorem A.1.** *Given a deep neural network $\mathcal{A}$ which consists of only convolution and linear layers. Let the network use one of $f(x) = \min\{x, 0\}$ (relu) or $f(x) = x$ (linear) as the activation function. Let the network be trained using the adjoint loss function as defined in Eqn. 3. Let $\mathbf{X}$ be the set of parameters of the network $\mathcal{A}$ which is shared across both the smaller and bigger networks. Let $\mathbf{Y}$ be the set of parameters of the bigger network not shared with the smaller network. Let $\mathbf{p}$ be the output of the larger network and let $\mathbf{q}$ be the output of the smaller network. Then, the adjoint loss function induces a data-dependent regularizer with the following properties.*

- *For all $x \in X$, the induced $L_2$ penalty is given by $\sum_i \mathbf{p}_i \big( \log' \mathbf{p}_i - \log' \mathbf{q}_i \big)^2$*

- *For all $y \in Y$, the induced $L_2$ penalty is given by $\sum_i \mathbf{p}_i \big( \log' \mathbf{p}_i \big)^2$*

*Proof.* We are interested in analyzing the regularizing behavior of the following loss function. $-y \log p + KL(p, q)$ $y$ is the ground truth label, $p$ is the output probability vector of the bigger network and $q$ is the output probability vector of the smaller network. Recall that the parameters of smaller network are shared across both. We will look at the second order taylor expansion for the kl-divergence term. This will give us insights into regularization behavior of the loss function.

Let $x$ be a parameter which is common across both the networks and $y$ be a parameter in the bigger network but not the smaller one.

$$D(x) = \sum_i p_i(x) \big( \log p_i(x) - \log q_i(x) \big) \text{ and } D(y) = \sum_i p_i(y) \big( \log p_i(y) - \log q_i \big)$$

For the parameter $y$, $q_i$ is a constant. Now, computing the first order derivative, we get that

$$D'(x) = \sum_i p_i'(x) \big( \log p_i(x) - \log q_i(x) \big) + p_i'(x) - \frac{q_i'(x) p_i(x)}{q_i(x)}$$

$$D'(y) = \sum_i p_i'(y) \big( \log p_i(y) - \log q_i \big) + p_i'(y)$$

Now, computing the second derivative for both the types of parameters, we get that

$$D''(x) = \sum_i p_i''(x) \big( \log p_i(x) - \log q_i(x) \big) + p_i'(x) \left( \frac{p_i'(x)}{p_i(x)} - \frac{q_i'(x)}{q_i(x)} \right) + p_i''(x)$$
$$- \frac{q_i(x) q_i'(x) p_i'(x) + q_i(x) q_i''(x) p_i(x) - q_i'(x) q_i'(x) p_i(x)}{q_i^2(x)}$$

$$D''(y) = \sum_i p_i''(y) \big( \log p_i(y) - \log q_i \big) + \frac{p_i'(y) p_i'(y)}{p_i(y)} + p_i''(y)$$

$$D''(x) = \sum_i \frac{p_i'(x) p_i'(x)}{p_i(x)} - \frac{2 p_i'(x) q_i'(x)}{q_i(x)} + \frac{q_i'(x) q_i'(x) p_i(x)}{q_i^2(x)} = \sum_i p_i(x) \left( \frac{p_i'(x)}{p_i(x)} - \frac{q_i'(x)}{q_i(x)} \right)^2$$

$$= \sum_i p_i (\log' p_i - \log' q_i)^2 \tag{4}$$

Similarly, for the parameters only in the bigger network, we get that

$$D''(y) = \sum_i \frac{p_i'(y)p_i'(y)}{p_i(y)} = \sum_i p_i (\log' p_i)^2 \tag{5}$$

Note that $y$ represents the over-parameterized weights of the model. The equations above show that the regularization imposed by the KL-divergence term on these parameters is such that if these parameters change a lot (on the log scale) then the penalty imposed on such parameters is more. Thus, the kl-divergence term encourages such parameters not to change by a lot. □

## B  DATA AUGMENTATION

We use different data-augmentation techniques for different datasets. Below are the details.

- *Cifar-10, Cifar-100* and *Oxford pets*
  We apply the following set of transforms for these datasets. (1) We flip the image with probability $0.5$. (2) With probability $0.75$, we rotate the image by degree $d$ chosen uniformly at random from $(-max_d, max_d)$. $max_d = 25.0$ for the cifar datasets and $max_d = 10.0$ for the pets dataset. (3) With probability $0.75$, we apply the contrast and brightness transforms. (4) With probability $0.75$, we apply the warp and zoom transforms. (5) We normalize the image by first dividing all the pixel values by $255$. and then subtracting the mean $[0.485, 0.456, 0.406]$ and dividing by the variance $[0.229, 0.224, 0.225]$.

- *Imagenet* and *Imagewoof*
  On these two datasets, we apply the following set fo transforms. (1) Random resize cropping - Crop a rectangular region with aspect ratio in $[3/4, 4/3]$ (selected uniformly at random) with area in $[0.08, 1.0]$ of the original area. (2) Flip the image horizontally with probability $0.5$. (3) Normalize the image by dividing all pixel values of $255.0$.

For cifar-10 and cifar-100, input size is $32 \times 32$ for all the other datasets, the input size is $224 \times 224$. The above transforms are applicable for the training dataset. For validation, we use center crop - select the center of the image with $85\%$ area, followed by a normalization step. Note that our data augmentation are not heavily optimized for accuracy. Rather our goal is to compare adjoint training with standard training. Hence, we use the same data augmentation steps for both the trainings. For standard training, our accuracies are still comparable to the accuracies reported in the literature on these datasets using the resnet18 and resnet50 architectures. However, the adjoint training methodology proposed in this paper outperforms the network trained in the standard way.

More details of the data-augmentation can be found in the code provided with the supplementary materials.

## C  RESNET ARCHITECTURE DIAGRAM

$$\rightarrow \underbrace{\begin{bmatrix} \text{conv}(3, 16) \\ \text{conv}(16, 64) \\ \text{conv}(64, 64) \end{bmatrix}}_{stem} \rightarrow MaxPool \rightarrow \underset{x3}{ResBlock(16, 64)} \rightarrow \underset{x4}{ResBlock(64, 128)} \rightarrow \underset{x6}{ResBlock(128, 256)}$$

$$\rightarrow \underset{x3}{ResBlock(256, 512)} \rightarrow AverageAdaptivePool \rightarrow Linear$$

Figure 5: Architecture diagram for resnet50 network used in this paper. $conv(ni, no)$ is a combination of convolution layer with $ni$ input and $no$ output channels followed by a batch norm and relu layer. The ResBlocks are as defined in Fig. 6.

$$\longrightarrow \begin{bmatrix} \text{conv(4ni, no)} \\ \text{conv(no, no)} \\ \text{conv(no, 4no)} \end{bmatrix} \longrightarrow \begin{bmatrix} \text{conv(4no, no)} \\ \text{conv(no, no)} \\ \text{conv(no, 4no)} \end{bmatrix}$$
$$x\,(l-1)$$

Figure 6: A ResBlock with $l$ layers and input $ni$ and output $no$.

The architecture for resnet50 is depicted in Figs. 5 and 6. Each $conv$ layer is actually a combination of three layers. A standard convolution layer followed by a batch normalization layer followed by a relu activation. The $ResBlock$ refers to the residual blocks in resnet architecture. Note that the skip connections are not shown in these diagrams. For a resnet18 architecture, each resblock is repeated twice instead of $3, 4, 6$ and $3$ times. Also, the resblock does not have a factor four in the convolution input and output.

For adjoint networks, the convolution parameters of all the last three resblocks are shared across both the original and the smaller architecture. Note that both the networks have different parameters for the batch-norm layers.

# D    DETAILED EXPERIMENTAL RESULTS

## D.1    EXPERIMENTS ON CIFAR-10

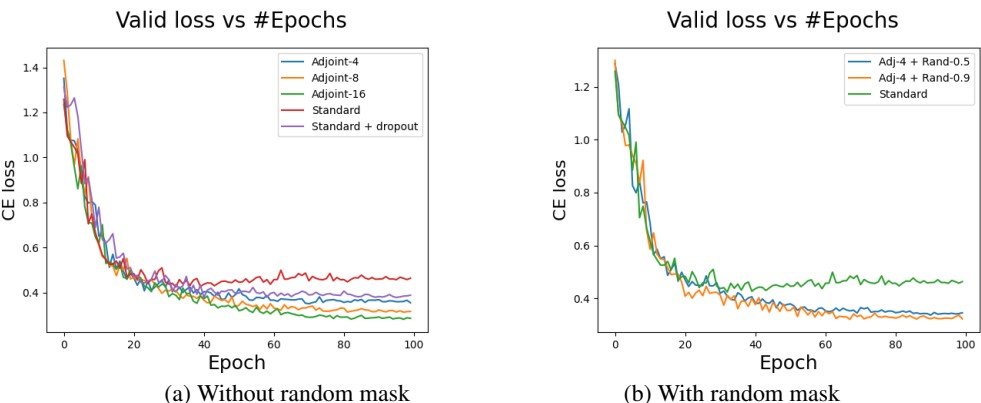

(a) Without random mask          (b) With random mask

Figure 7: Validation loss for the various training paradigms for resnet50 trained on Cifar-10. All the adjoint plots correspond to the bigger network.

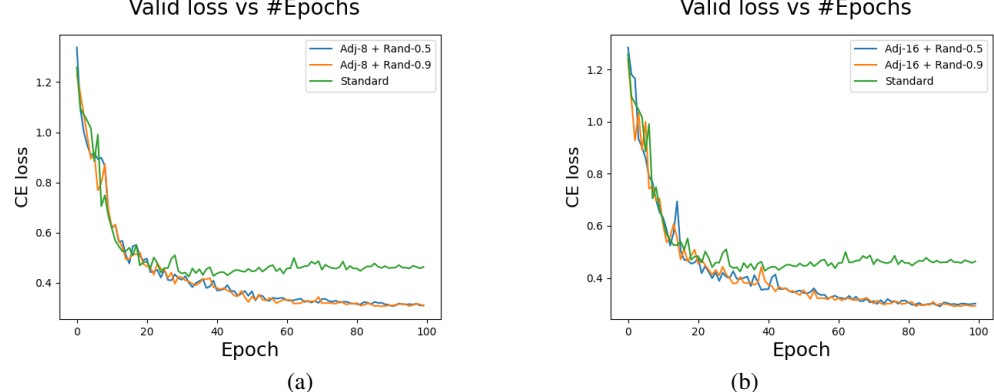

(a)                          (b)

Figure 8: Validation loss for the various training paradigms for resnet50 trained on Cifar-10. Validation loss for the various training paradigms for resnet50 trained on Cifar-10. All the adjoint plots correspond to the bigger network.

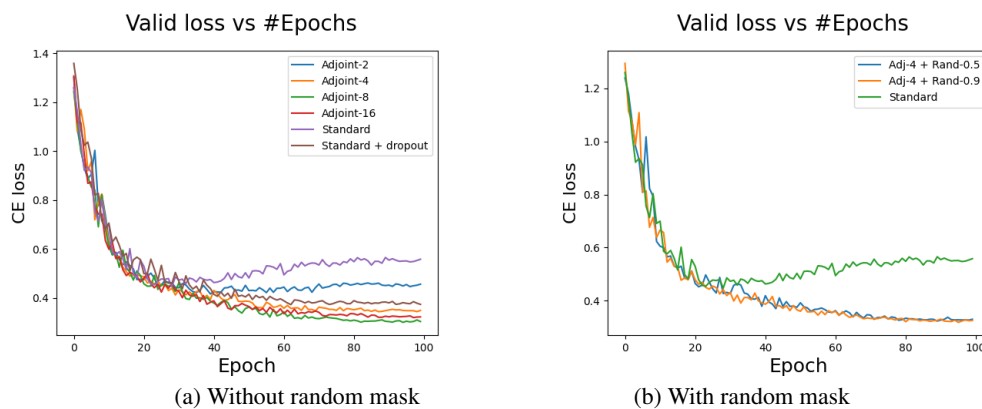

(a) Without random mask         (b) With random mask

Figure 9: Validation loss for the various training paradigms for resnet18 trained on Cifar-10. All the adjoint plots correspond to the bigger network.

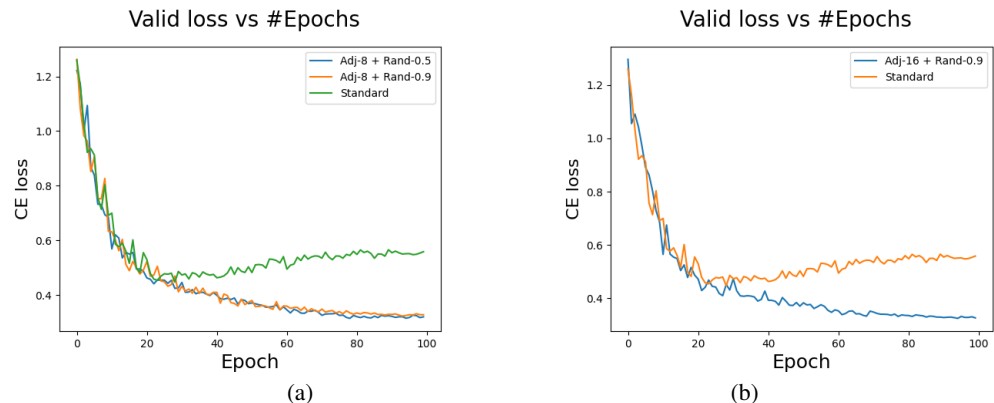

(a)             (b)

Figure 10: Validation loss for the various training paradigms for resnet18 trained on Cifar-10. All the adjoint plots correspond to the bigger network.

| Training with resnet50 on cifar-10 | | | | | |
|---|---|---|---|---|---|
| Training paradigm | Masking matrix ($M$) | top-1 full | top-5 full | top-1 small | top-5 small |
| Standard-full | | 90.25 | 99.67 | | |
| Standard-full + dropout (0.75) | | 90.11 | 99.68 | | |
| Adjoint-4 | $a_4$ | 90.83 | 99.7 | 90.25 | 99.6 |
| Adjoint-8 | $a_8$ | 91.06 | 99.67 | 89.77 | 99.63 |
| Adjoint-16 | $a_16$ | 91.34 | 99.71 | 89.88 | 99.61 |
| Adjoint-4 | $a_4 * r_{0.5}$ | 90.56 | 99.75 | 90.13 | 99.73 |
| Adjoint-4 | $a_4 * r_{0.9}$ | 91.12 | 99.69 | 90.1 | 99.7 |
| Adjoint-8 | $a_8 * r_{0.5}$ | 91.01 | 99.77 | 89.64 | 99.71 |
| Adjoint-8 | $a_8 * r_{0.9}$ | 91.31 | 99.73 | 90.05 | 99.61 |
| Adjoint-16 | $a_{16} * r_{0.5}$ | 91.13 | 99.59 | 89.81 | 99.63 |
| Adjoint-16 | $a_{16} * r_{0.9}$ | 91.2 | 99.68 | 89.42 | 99.61 |

Table 4: Accuracy for the various training paradigms for resnet50 trained on Cifar-10

| Training with resnet18 on cifar-10 | | | | | | |
|---|---|---|---|---|---|---|
| Training paradigm | Masking matrix ($M$) | top-1 full | top-5 full | top-1 small | top-5 small |
| Standard-full | | 88.17 | 99.41 | | |
| Standard-full + dropout (0.5) | | 89.04 | 99.55 | | |
| Adjoint-2 | $a_2$ | 88.75 | 99.54 | 88.06 | 99.41 |
| Adjoint-4 | $a_4$ | 89.8 | 99.73 | 88.74 | 99.64 |
| Adjoint-8 | $a_8$ | 90.26 | 99.72 | 88.62 | 99.62 |
| Adjoint-16 | $a_{16}$ | 89.58 | 99.61 | 87.94 | 99.57 |
| Adjoint-4 | $a_4 * r_{0.9}$ | 89.88 | 99.65 | 88.59 | 99.53 |
| Adjoint-4 | $a_4 * r_{0.9}$ | 89.75 | 99.56 | 88.34 | 99.48 |
| Adjoint-8 | $a_8 * r_{0.5}$ | 89.96 | 99.61 | 87.85 | 99.52 |
| Adjoint-8 | $a_8 * r_{0.9}$ | 89.4 | 99.56 | 87.61 | 99.51 |
| Adjoint-16 | $a_{16} * r_{0.9}$ | 89.66 | 99.52 | 87.67 | 99.35 |

Table 5: Accuracy for the various training paradigms for resnet18 trained on Cifar-10

## D.2 EXPERIMENTS ON CIFAR-100

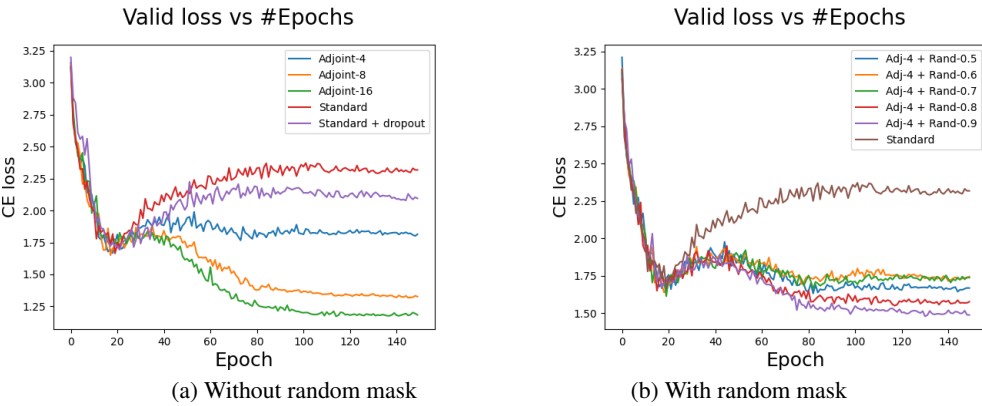

(a) Without random mask  (b) With random mask

Figure 11: Validation loss for the various training paradigms for resnet50 trained on Cifar-100. All the adjoint plots correspond to the bigger network.

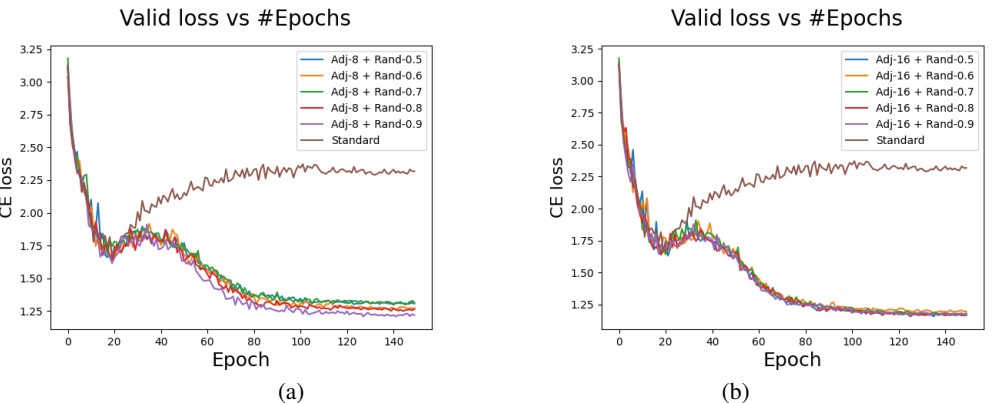

(a)  (b)

Figure 12: Validation loss for the various training paradigms for resnet50 trained on Cifar-100. All the adjoint plots correspond to the bigger network.

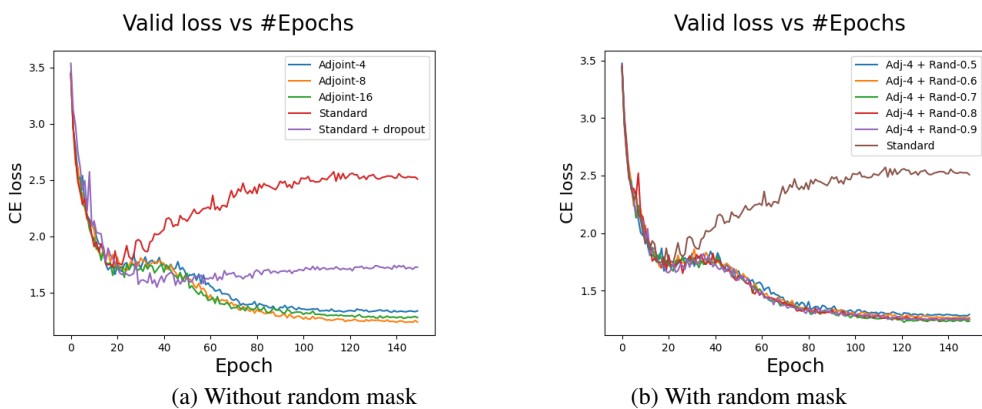

(a) Without random mask        (b) With random mask

Figure 13: Validation loss for the various training paradigms for resnet18 trained on Cifar-100. All the adjoint plots correspond to the bigger network.

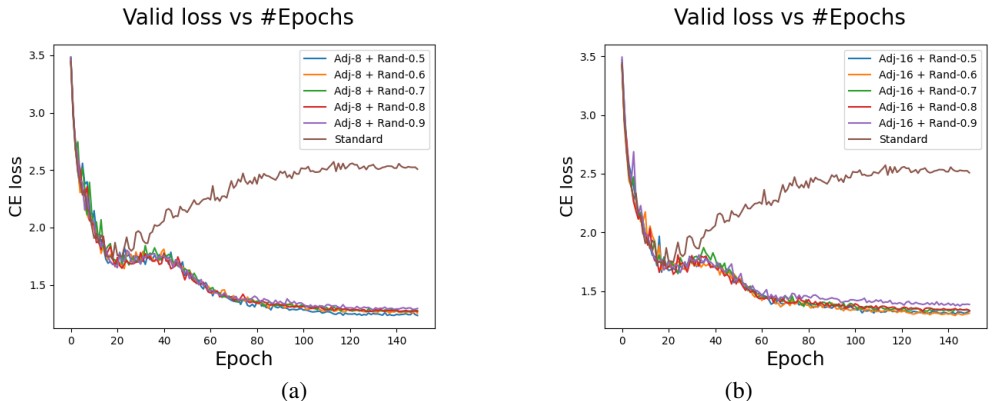

(a)        (b)

Figure 14: Validation loss for the various training paradigms for resnet18 trained on Cifar-100. All the adjoint plots correspond to the bigger network.

| Training with resnet50 on cifar-100 | | | | | | |
|---|---|---|---|---|---|---|
| Training paradigm | Masking matrix ($M$) | top-1 full | top-5 full | top-1 small | top-5 small |
| Standard-full | | 65.31 | 87.35 | | |
| Standard-full + dropout (0.75) | | 66.66 | 88.24 | | |
| Adjoint-4 | $a_4$ | 67.15 | 89.04 | 64.73 | 87.6 |
| Adjoint-8 | $a_8$ | 68.75 | 90.79 | 65.37 | 88.88 |
| Adjoint-16 | $a_16$ | 69.07 | 91.11 | 65.11 | 89.08 |
| Adjoint-4 | $a_4 * r_{0.5}$ | 67.47 | 89.78 | 64.71 | 88.12 |
| Adjoint-4 | $a_4 * r_{0.6}$ | 67.45 | 89.22 | 64.56 | 87.56 |
| Adjoint-4 | $a_4 * r_{0.7}$ | 67.35 | 89.69 | 64.26 | 87.84 |
| Adjoint-4 | $a_4 * r_{0.8}$ | 67.82 | 89.71 | 64.71 | 88.13 |
| Adjoint-4 | $a_4 * r_{0.9}$ | 67.57 | 89.99 | 65.04 | 88.24 |
| Adjoint-8 | $a_8 * r_{0.5}$ | 68.7 | 90.57 | 65.35 | 89.05 |
| Adjoint-8 | $a_8 * r_{0.6}$ | 68.86 | 90.68 | 65.47 | 89.27 |
| Adjoint-8 | $a_8 * r_{0.7}$ | 68.57 | 90.71 | 64.76 | 88.6 |
| Adjoint-8 | $a_8 * r_{0.8}$ | 67.82 | 89.84 | 64.76 | 88.01 |
| Adjoint-8 | $a_8 * r_{0.9}$ | 69.09 | 90.99 | 65.11 | 89.13 |
| Adjoint-16 | $a_{16} * r_{0.5}$ | 68.59 | 91.13 | 64.53 | 89.1 |
| Adjoint-16 | $a_{16} * r_{0.6}$ | 68.04 | 90.79 | 64.55 | 88.66 |
| Adjoint-16 | $a_{16} * r_{0.7}$ | 68.85 | 91.03 | 64.76 | 88.99 |
| Adjoint-16 | $a_{16} * r_{0.8}$ | 68.52 | 90.93 | 64.35 | 89.29 |
| Adjoint-16 | $a_{16} * r_{0.9}$ | 68.89 | 90.79 | 64.54 | 89.48 |

Table 6: Accuracy for the various training paradigms for resnet50 trained on Cifar-100

| Training with resnet18 on cifar-100 | | | | | | |
|---|---|---|---|---|---|---|
| Training paradigm | Masking matrix ($M$) | top-1 full | top-5 full | top-1 small | top-5 small |
| Standard-full | | 61.39 | 85.38 | | |
| Standard-full + dropout (0.5) | | 64.3 | 88.13 | | |
| Adjoint-4 | $a_4$ | 66.84 | 89.2 | 62.73 | 87.4 |
| Adjoint-8 | $a_8$ | 66.09 | 89.43 | 61.38 | 87.12 |
| Adjoint-16 | $a_{16}$ | 64.48 | 89.02 | 57.84 | 85.24 |
| Adjoint-4 | $a_4 * r_{0.5}$ | 66.54 | 89.56 | 62.64 | 87.38 |
| Adjoint-4 | $a_4 * r_{0.6}$ | 65.81 | 89.49 | 62.26 | 87.52 |
| Adjoint-4 | $a_4 * r_{0.7}$ | 65.88 | 89.75 | 61.7 | 87.53 |
| Adjoint-4 | $a_4 * r_{0.8}$ | 66.13 | 89.55 | 61.91 | 87.08 |
| Adjoint-4 | $a_4 * r_{0.9}$ | 66.29 | 89.55 | 61.42 | 86.89 |
| Adjoint-8 | $a_8 * r_{0.5}$ | 65.63 | 89.28 | 60.64 | 86.79 |
| Adjoint-8 | $a_8 * r_{0.6}$ | 65.14 | 89.08 | 60.08 | 86.39 |
| Adjoint-8 | $a_8 * r_{0.7}$ | 64.49 | 89.03 | 58.93 | 85.63 |
| Adjoint-8 | $a_8 * r_{0.8}$ | 64.88 | 89.17 | 59.45 | 86.09 |
| Adjoint-8 | $a_8 * r_{0.9}$ | 64.34 | 88.54 | 58.17 | 85.15 |
| Adjoint-16 | $a_{16} * r_{0.5}$ | 63.66 | 88.38 | 56.22 | 84.68 |
| Adjoint-16 | $a_{16} * r_{0.6}$ | 64.07 | 88.27 | 56.83 | 84.22 |
| Adjoint-16 | $a_{16} * r_{0.7}$ | 63.14 | 88.02 | 54.78 | 83.65 |
| Adjoint-16 | $a_{16} * r_{0.8}$ | 63.26 | 87.85 | 54.52 | 83.2 |
| Adjoint-16 | $a_{16} * r_{0.9}$ | 62.32 | 87.37 | 52.88 | 82.13 |

Table 7: Accuracy for the various training paradigms for resnet18 trained on Cifar-100

### D.3 EXPERIMENTS ON IMAGENET

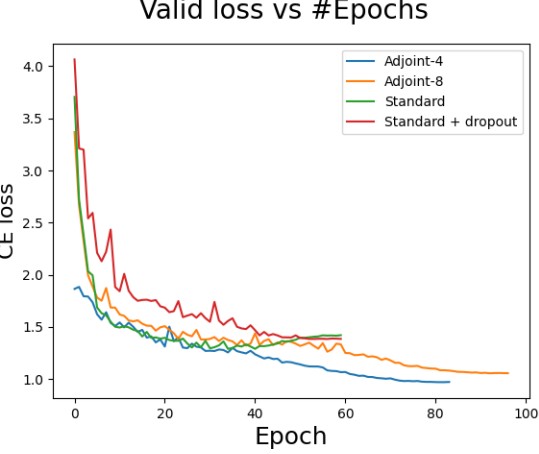

Figure 15: Validation loss for the various training paradigms for resnet50 trained on Imagenet. All the adjoint plots correspond to the bigger network.

| Training with resnet50 on imagenet | | | | | | |
|---|---|---|---|---|---|---|
| Training paradigm | Masking matrix ($M$) | top-1 full | top-5 full | top-1 small | top-5 small |
| Standard-full | | 73.41 | 91.07 | | |
| Standard-full + dropout (0.75) | | 69.54 | 88.88 | | |
| Adjoint-4 | $a_4$ | 75.84 | 92.77 | 71.84 | 90.42 |
| Adjoint-8 | $a_8$ | 73.46 | 91.52 | 64.7 | 85.96 |

Table 8: Accuracy for the various training paradigms for resnet50 trained on imagenet

### D.4 EXPERIMENTS ON IMAGEWOOF

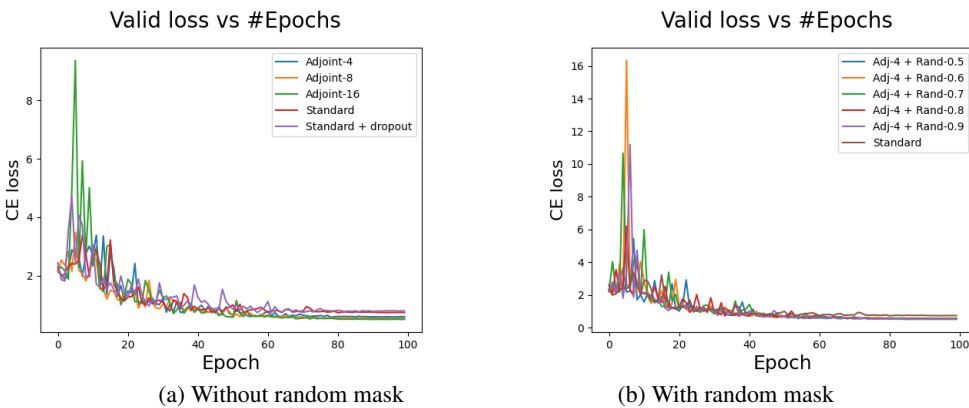

(a) Without random mask

(b) With random mask

Figure 16: Validation loss for the various training paradigms for resnet50 trained on imagewoof. All the adjoint plots correspond to the bigger network.

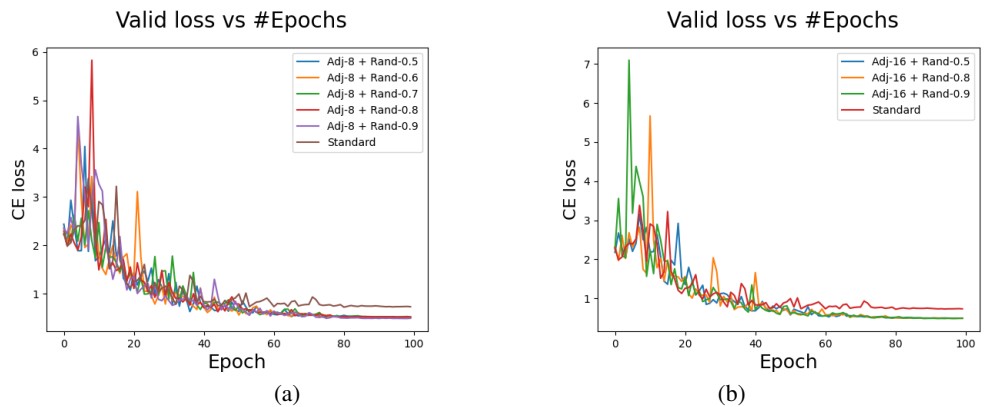

(a)

(b)

Figure 17: Validation loss for the various training paradigms for resnet50 trained on imagewoof. All the adjoint plots correspond to the bigger network.

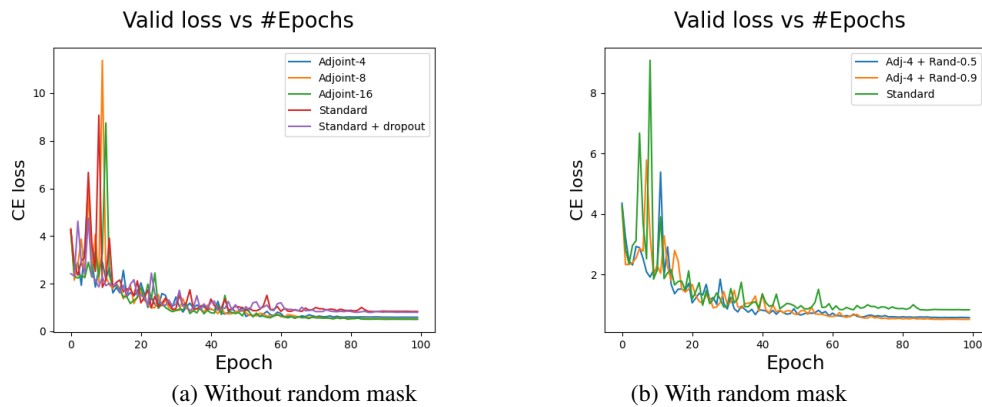

(a) Without random mask  (b) With random mask

Figure 18: Validation loss for the various training paradigms for resnet18 trained on imagewoof. All the adjoint plots correspond to the bigger network.

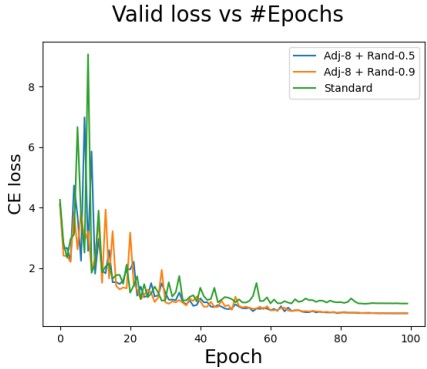

Figure 19: Validation loss for the various training paradigms for resnet18 trained on imagewoof. All the adjoint plots correspond to the bigger network.

| Training with resnet18 on imagewoof | | | | | |
|---|---|---|---|---|---|
| Training paradigm | Masking matrix ($M$) | top-1 full | top-5 full | top-1 small | top-5 small |
| Standard-full | | 83.35 | 98.3 | | |
| Standard-full + dropout (0.5) | | 82.1 | 98.09 | | |
| Adjoint-4 | $a_4$ | 84.16 | 98.38 | 82.96 | 98.3 |
| Adjoint-8 | $a_8$ | 84.11 | 98.54 | 81.25 | 98.02 |
| Adjoint-16 | $a_{16}$ | 83.59 | 98.43 | 79.37 | 97.78 |
| Adjoint-4 | $a_4 * r_{0.5}$ | 84.14 | 98.48 | 82.81 | 97.99 |
| Adjoint-4 | $a_4 * r_{0.9}$ | 83.82 | 98.17 | 80.54 | 98.02 |
| Adjoint-8 | $a_8 * r_{0.5}$ | 84.27 | 98.69 | 81.69 | 98.28 |
| Adjoint-8 | $a_8 * r_{0.9}$ | 84.06 | 98.41 | 78.8 | 97.78 |

Table 9: Accuracy for the various training paradigms for resnet18 trained on imagewoof

| Training with resnet50 on imagewoof | | | | | | |
|---|---|---|---|---|---|---|
| Training paradigm | Masking matrix ($M$) | top-1 full | top-5 full | top-1 small | top-5 small |
| Standard-full | | 85.2 | 98.4 | | |
| Standard-full + dropout (0.75) | | 82.76 | 98.38 | | |
| Adjoint-4 | $a_4$ | 85.52 | 98.3 | 85.26 | 98.17 |
| Adjoint-8 | $a_8$ | 85.62 | 98.59 | 84.11 | 98.35 |
| Adjoint-16 | $a_{16}$ | 85.28 | 98.54 | 82.94 | 98.12 |
| Adjoint-4 | $a_4 * r0.5$ | 85.75 | 98.48 | 85 | 98.48 |
| Adjoint-4 | $a_4 * r0.6$ | 85.75 | 98.69 | 85.15 | 98.54 |
| Adjoint-4 | $a_4 * r0.7$ | 85.31 | 98.38 | 84.5 | 98.35 |
| Adjoint-4 | $a_4 * r0.8$ | 85.65 | 98.64 | 84.76 | 98.43 |
| Adjoint-4 | $a_4 * r0.9$ | 86.3 | 98.59 | 84.27 | 98.48 |
| Adjoint-8 | $a_8 * r0.5$ | 85.72 | 98.67 | 83.95 | 98.28 |
| Adjoint-8 | $a_8 * r0.6$ | 85.62 | 98.64 | 84.01 | 98.41 |
| Adjoint-8 | $a_8 * r0.7$ | 85.62 | 98.48 | 83.3 | 98.35 |
| Adjoint-8 | $a_8 * r0.8$ | 85.49 | 98.51 | 82.52 | 98.38 |
| Adjoint-8 | $a_8 * r0.9$ | 85.65 | 98.75 | 83.41 | 98.33 |
| Adjoint-16 | $a_{16} * r0.5$ | 86.01 | 98.56 | 83.09 | 98.25 |
| Adjoint-16 | $a_{16} * r0.8$ | 85.39 | 98.61 | 82.73 | 98.12 |
| Adjoint-16 | $a_{16} * r0.9$ | 85.33 | 98.67 | 82.26 | 98.38 |

Table 10: Accuracy for the various training paradigms for resnet50 trained on imagewoof

### D.5  EXPERIMENTS ON OXFORD-PETS

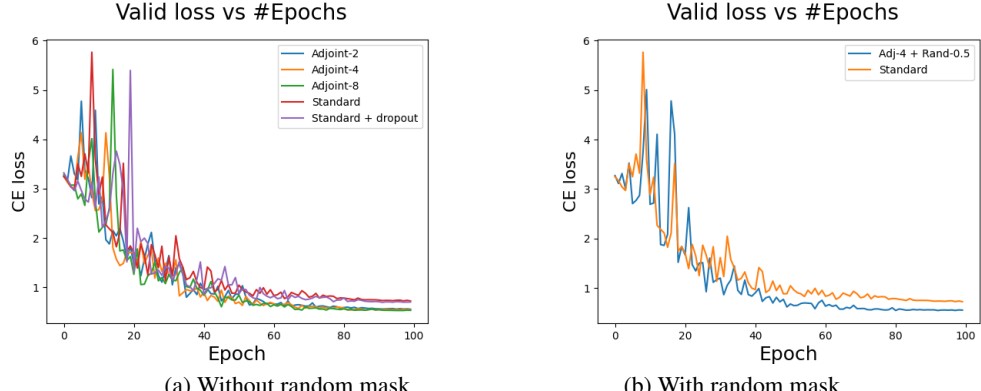

(a) Without random mask

(b) With random mask

Figure 20: Validation loss for the various training paradigms for resnet50 trained on oxford-pets All the adjoint plots correspond to the bigger network.

Figure 21: Validation loss for the various training paradigms for resnet50 trained on oxford-pets. All the adjoint plots correspond to the bigger network.

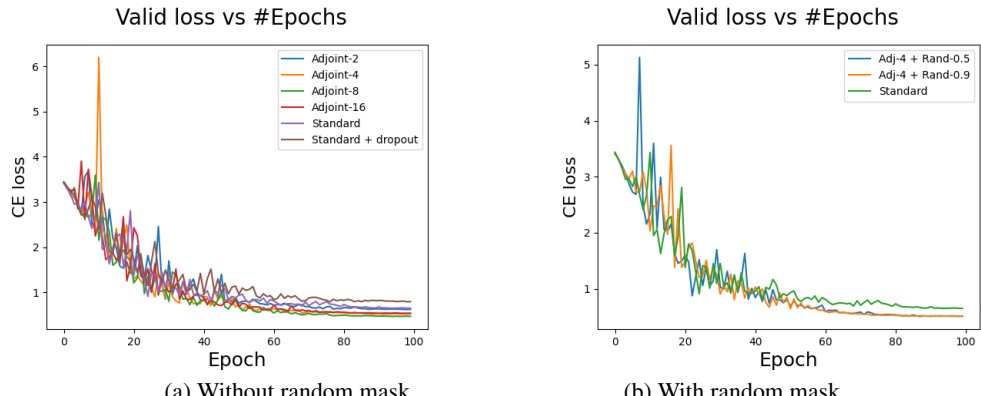

(a) Without random mask
(b) With random mask

Figure 22: Validation loss for the various training paradigms for resnet18 trained on oxford-pets. All the adjoint plots correspond to the bigger network.

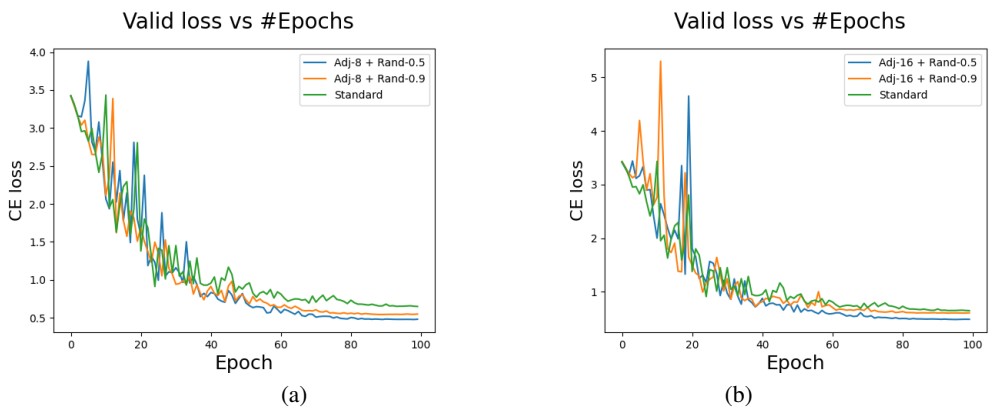

(a)
(b)

Figure 23: Validation loss for the various training paradigms for resnet18 trained on oxford-pets. All the adjoint plots correspond to the bigger network.

| Training with resnet50 on oxford-pets | | | | | | |
|---|---|---|---|---|---|---|
| Training paradigm | Masking matrix ($M$) | top-1 full | top-5 full | top-1 small | top-5 small |
| Standard-full | | 85.31 | 97.63 | | |
| Standard-full + dropout (0.5) | | 84.7 | 98.1 | | |
| Adjoint-2 | $a_2$ | 87 | 98.03 | 86 | 98.1 |
| Adjoint-4 | $a_4$ | 86.73 | 98.24 | 85.58 | 98.03 |
| Adjoint-8 | $a_8$ | 85.85 | 98.24 | 84.03 | 98.1 |
| Adjoint-2 | $a_2 * r_{0.5}$ | 86.6 | 97.22 | 85.85 | 97.36 |
| Adjoint-2 | $a_2 * r_{0.9}$ | 85.52 | 98.17 | 85.58 | 97.69 |
| Adjoint-4 | $a_4 * r_{0.5}$ | 86.12 | 97.9 | 85.38 | 97.56 |
| Adjoint-4 | $a_4 * r_{0.9}$ | 85.25 | 97.63 | 83.69 | 97.69 |
| Adjoint-8 | $a_8 * r_{0.5}$ | 86.33 | 98.37 | 84.64 | 97.63 |
| Adjoint-8 | $a_8 * r_{0.5}$ | 85.85 | 98.37 | 81.93 | 98.51 |

Table 11: Accuracy for the various training paradigms for resnet50 trained on oxford-pets

| Training with resnet18 on oxford-pets | | | | | | |
|---|---|---|---|---|---|---|
| Training paradigm | Masking matrix ($M$) | top-1 full | top-5 full | top-1 small | top-5 small |
| Standard-full | | 84.84 | 97.76 | | |
| Standard-full + dropout (0.5) | | 82.61 | 97.09 | | |
| Adjoint-2 | $a_2$ | 84.5 | 97.56 | 83.89 | 97.36 |
| Adjoint-4 | $a_4$ | 86.33 | 98.3 | 83.08 | 97.76 |
| Adjoint-8 | $a_8$ | 85.52 | 98.71 | 82.61 | 97.97 |
| Adjoint-2 | $a_2 * r_{0.5}$ | 84.91 | 97.69 | 83.62 | 97.63 |
| Adjoint-2 | $a_2 * r_{0.9}$ | 83.76 | 97.76 | 80.71 | 97.22 |
| Adjoint-4 | $a_4 * r_{0.5}$ | 85.65 | 97.9 | 82.74 | 97.83 |
| Adjoint-4 | $a_4 * r_{0.9}$ | 84.84 | 97.83 | 80.51 | 97.09 |
| Adjoint-8 | $a_8 * r_{0.5}$ | 85.31 | 97.9 | 82.07 | 97.63 |
| Adjoint-8 | $a_8 * r_{0.9}$ | 83.33 | 97.9 | 76.72 | 96.48 |
| Adjoint-16 | $a_16 * r_{0.5}$ | 84.5 | 97.83 | 77.46 | 96.75 |
| Adjoint-16 | $a_16 * r_{0.9}$ | 82.27 | 97.76 | 72.05 | 95.26 |

Table 12: Accuracy for the various training paradigms for resnet18 trained on oxford-pets

# E  CHOOSING THE REGULARIZATION FUNCTION

Finally, we compare different choices of regularization function for the adjoint loss (Eqn. 3). For all the previous experiments, we use the 'quadratic' function $\lambda(t) = \min\{4t^2, 1\}$. In this section, we fix the architecture, dataset and mask matrix as resnet18, cifar100 and Adj-4 respectively and vary the regularization function. We look at different functions which includes exponential and trigonometric

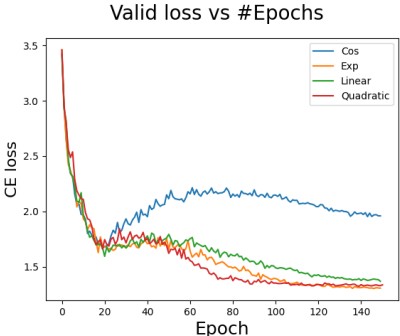

Figure 24: Validation cross entropy loss for various regularization functions. The networks were trained using Adj-4 mask matrix on cifar-100 using resnet-18.

| Adjoint trained with different regularization functions | | |
|---|---|---|
| Regularization function $\lambda(t)$ | top-1 full | top-1 small |
| $1 - \cos(t)$ | $-2.86$ | $-3.14$ |
| $t$ | $-0.1$ | $-0.15$ |
| $\min\{4t^2, 1\}$ | $0.00$ | $0.00$ |
| $\exp(t) - 1$ | $-0.37$ | $+0.38$ |

Table 13: The effect of training with different regularization functions on the top-1 accuracies of the bigger and the smaller networks. The quadratic function $\min\{4t^2, 1\}$ is used as the base for comparison.

functions. Table 13 and Fig. 24 both show the same trend. The cos function performs the worst while the rest have similar performance. We conjecture that any function that is close to zero for $t \leftarrow 0$ and grows to one eventually should be a reasonable choice for the regularization function. Note that throughout our discussion, we have used $\lambda(t) = c \min\{4t^2, 1\}$ with $c = 1$. Depending on the dataset, other values of $c$ maybe more appropriate.

