# OpenReview forum: "Better Together: Resnet-50 accuracy with $13 \times $ fewer parameters and at $3 \times $ speed"
_ICLR.cc/2021/Conference — Reject_

### Official Review · AnonReviewer4 · 2020-10-23
**A practical training approach that can regularize and compress any CNN-based neural architecture.**

**Rating:** 6
**Confidence:** 3

**Review:**

The paper proposes “Adjoined networks” as a training approach that can regularize and compress any CNN-based neural architecture in a manner of one-shot learning paradigm. The proposed technique deals with the main issues of current compression approaches. That is (1) Requiring the availability of special hardware to support fast inference and (2) Requiring large training time as they alternate between pruning and fine-tuning.

1)	The section 2 of Adjoined networks is not clearly illustrated. The author says that the small network selects a fraction of the convolution filters of the large one as its filters. However, the channel of random selection filters may be inconsistent with the feature maps of the deeper layers. How to implement the convolution in small network needs to be further illustrated?
2)	The author needs to clarify whether the M is optional expect the setting as 16/8/4/2 in the work. A binary mask matrix M is used to compress parameters. However, the M is fixed before training and is not learnt. In the experiments, M is such a matrix that the first 16/8/4/2 rows are all 1 and the rest 0. In table 9, with M zeros-outing parameters incrementally, the results do not follow the same trendy.
3)	The proposed “Adjoint loss” is similar with the loss of existing knowledge distillation architectures. The author needs to illustrate the difference between them.

---

> ### Author Response · Authors · 2020-11-25
> **Clarifications**
>
> 1 and 2 - Let the input to the network be of size $w \times h \times c_{in}$. For the first convolution layer, both the smaller and larger network get the same input. The weights of the first layer have shape $k \times k \times c_{in} \times c_{out}$. Hence, the output for the larger network has shape $w' \times h' \times c_{out}$ and the smaller network has shape $w' \times h' \times c_{out} / \alpha$. Here, $\alpha$ is a tunable hyper-parameter and can be $16, 8, 4, 2$ or any other number.  The first output corresponds to $conv(inp, W)$ and the second corresponds to $conv(inp, W*M)$.
>
> While, the input for the first convolution layer is same for both the networks, the input for the two networks may diverge for the subsequent layers.  For example, for the second layer, the inputs have shape  $w \times h \times c_{in}$ and $w \times h \times c_{in}/\alpha$. The convolution layer has weights of shape $k \times k \times c_{in} \times c_{out}$ and $k \times k \times c_{in} /\alpha \times c_{out}/\alpha$ respectively. Thus, we see that there is no conflict in shapes. Also, we see that the number of parameters reduces approximately quadratically with $\alpha$.
>
> 3 - The reviewer correctly points out that our loss function has some similarities to the knowledge-distillation loss. However, our loss function is also "time-dependent" and tunable. $L = -y \log p + \lambda(t) KL(p, q)$ where $\lambda(t) = \min 4t^2, 1$.
> By choosing the function $\lambda$, we can choose to which rate we want the smaller network to learn from the larger network.

---

### Official Review · AnonReviewer3 · 2020-10-27
**Review for Better Together: Resnet-50 accuracy with  13x fewer parameters and at  3x speed**

**Rating:** 4
**Confidence:** 4

**Review:**

This paper introduces a training policy for jointly learning parameters of both supernets (or big teacher models) and subnets (or small student models). The student models have the similar architectures with teacher models, but have different numbers of convolution filters. For jointly training teacher models and student models, an adjoint loss consisted of cross-entropy loss and KL-div is defined. The experiments are conducted on several image benchmarks using ResNet-18 and ResNet-50.

Strengths:
+: The experimental results show the proposed method can significantly reduce model complexity with slight performance loss.
+: The proposed method seems simple and easy to implement.


Weaknesses:
-: Lacking more clarification on significances of this work.
(1) No state-of-the-art model compression method is compared for verifying the effectiveness of the proposed method. For filter pruning, many recently proposed methods [r1, r2, r3, r4] are presented to compress different sizes of CNN models (e.g., VGG, ResNet18, ResNet34, ResNet50, ResNet101, ResNet152) on various image benchmarks (e.g., CIFAR-10, CIFAR-100 and ImageNet). The corresponding discussion and comparison are missing in this work.
[r1] Discrimination-aware Channel Pruning for Deep Neural Networks. NIPS, 2018.
[r2] Gate Decorator: Global Filter Pruning Method for Accelerating Deep Convolutional Neural Networks. NIPS, 2019.
[r3] Channel Pruning via Automatic Structure Search. IJCAI, 2020.
[r4] HRank: Filter Pruning using High-Rank Feature Map. CVPR, 2020.

(2) The idea of training subnets from supernets shares similar philosophy with one-shot NAS. Particularly, one-shot NAS based on knowledge distillation has been studied [r5], which supernets and subnets are jointly trained. Meanwhile, [r5] also shows the searched small student models can achieve comparable or better performance than big teacher models. Therefore, what are advantages of the proposed method over [r5]?
[r5] Neural Architecture Search by Block-wisely Distilling Architecture Knowledge. CVPR, 2020.

(3) I wonder that why the results (73.41%) of ResNet-50 on ImageNet in this paper is inferior to those of other works (~75.1%). Besides, why the authors did not report the results of ResNet-18 on ImageNet?

(4) For fully verifying the effectiveness of the proposed method, the authors would better report the results using more CNN models, rather than those on various benchmarks. In particular, Imagewoof shares similar philosophy with ImageNet.

-: The writing is too colloquial, and there are too many typos.
(1) $13.7x$ -> $13.7 \times$
(2) kl -> KL
(3) the the smaller -> the smaller
(4) dropouts -> Dropout

-: I wonder that the meaning of Table 3. For the common settings, ResNet does not adopt Dropout. Therefore, is it fair or necessary for comparing ResNet with Dropout?

-: How about the transfer (generalization) ability of the proposed adjoined networks?

---

> ### Author Response · Authors · 2020-11-25
> **Response**
>
> We thank the reviewer for the comments. We hope that the reviewer will reconsider the evaluation after our clarifications.
>
> $\textbf{Comparison against other works}$
>
> We thank the reviewer for providing the relevant references. We added a section in the current draft which compares our method against other state of the art methods. The following table gives the details of the comparison. The results show that our method significantly outperforms all the other methods. While all other methods achieve a compression of about 2x or 3x our method achieves a compression of about 14x. One some other datasets like CIFAR-100, we even achieve compression of 99x which is order of magnitude better than other methods referenced.
>
> Method, Top-1 accuracy difference, Inference speed-up, Parameter reduction, Reference
>
> ABCPruner, 2.15, 2.18X, 2.17X, https://arxiv.org/pdf/2001.08565.pdf
>
> GBN, 0.67, 2.22X, 2.14X, https://arxiv.org/pdf/1909.08174.pdf
>
> DCP, 1.06, 2.25X, 2.06X, https://papers.nips.cc/paper/2018/file/55a7cf9c71f1c9c495413f934dd1a158-Paper.pdf
>
> HRank, 4.17, 2.63X, 1.86X, https://arxiv.org/pdf/2002.10179.pdf
>
> HRank, 7.05, 4.17X, 3.08X, https://arxiv.org/pdf/2002.10179.pdf
>
> Adjoint Network, 1.57, 3.1X, 13.7X, Current work
>
> The table above shows the difference in the top-1 accuracy between the standard network vs the network trained and pruned/compressed using the method referenced on the Imagenet dataset using the ResNet-50 architecture. We have added a experimental section in the main version of the paper.
>
>
> $\textbf{NAS}$
>
> We thank the reviewer for pointing out this reference. The reviewer is correct that philosophically [r5] also uses the same idea of jointly training the bigger and the smaller network together. However, the search is over architectures with a particular format; V1, V2, V4 and IT resembling the organization of the human visual cortex.  In contrast our method is plug-and-play. The design of the adjoint paradigm makes it easy to "swap out" any standard convolution layer and replace it with the adjoint convolution layer. There are also differences in the loss function.  Our loss function is also "time-dependent" and tunable $L = -y \log p + \lambda(t) KL(p, q)$. By choosing the $\lambda$ function, we can choose to which rate we want the smaller network to learn from the larger network.
>
>
> $\textbf{Imagenet}$
>
> We trained the resnet-50 architecture on imagenet using the standard code available on pytorch github repo. https://github.com/pytorch/examples/ blob/master/imagenet/main.py  . We agree that different papers report different accuracies (in the range 73-75%) for the resnet50 on imagenet. For multi-gpu training accuracy in the same range (~73.5%) has also been reported by the official nvidia blog. https://developer.nvidia.com/blog/mixed-precision-training-deep-neural-networks/.
>
> The project has been executed by two students who were severely budget constrained. We had to spend money out of our own pockets to do multi-GPU training on Imagenet. Hence, we had to be extremely selective on the choice and number of experiments we could do on the Imagenet dataset. We apologize that we could not execute the same experiment for resnet-18 as well.
>
>
> $\textbf{Formatting and other issues}$
>
> We have addressed the typos as pointed out by the reviewer.
>
> We agree with the reviewer that Imagewoof shares several similarities with Imagenet. But we have compared our method on various other datasets like CIFAR-10 and CIFAR-100, MNIST as well as Pets dataset. The choice of datasets is fairly typical of other papers. We have identified the following areas as avenues for future work. Comparison with other CNN models used in object detection, pose estimation and other vision models. In a preliminary work, we have seen 3.5x parameter reduction in a pose estimation model. However, we feel that including these comparisons is beyond the scope of this paper and will be covered in the follow-up paper that we are working on.
>
> The reviewer correctly points out the dropouts are not used with resnet architectures. We agree that the comparison is not entirely fair. In dropouts also the output is computed only a fraction of the weights (chosen at random). In adjoint paradigm as well the output of the smaller network is computed using only a fraction of weights. Hence, there is some conceptual similarity between the two training regimes and we included a section comparing the two.
>
> The generalization ability of adjoint network can be seen from Tables 1 and 2 and Figure 3. This shows that the network trained in the adjoint fashion has better regularization and better validation accuracy than the same network trained in the standard fashion.

---

### Official Review · AnonReviewer2 · 2020-10-28
**New approach for teacher-student joint training**

**Rating:** 5
**Confidence:** 3

**Review:**

The authors propose a method for training two networks jointly. This resembles the teacher-student approach where the teacher (large) network is trained first and used as an instructor for training the student network (smaller). The proposed paper implements the training of the two networks in a joint procedure and proposes a new kind of loss function "adjoint loss" that consists of two terms: (1) the prediction loss of the teacher network, and (2) the Kullback-Leibler divergence between the predictions of the student and the teacher. Moreover, the weights of the student are a subset of those of the teacher.

The method is similar to classical knowledge distillation approach (student-teacher paradigm by Hinton et al 2015). The key difference seems to be in weight sharing across the two networks. This idea is partly novel, but should refer to the 2018 paper "Rocket Launching: A Universal and Efﬁcient Framework for Training Well-Performing Light Net" by Zhou et al, who also consider weight sharing but use a different (but similar) loss function.

The results should be clarified. The impression is that the comparison is primarily against the conventional teacher-student setting (Hinton, 2015). This comparison is not relevant, as there are over 4500 citations to the original work. Also in more general terms, the proposed method should better reflect the already lengthy history of knowledge distillation, and position the current work better in the landscape.

---

> ### Author Response · Authors · 2020-11-25
> **Differences with teacher student training**
>
> We thank the reviewer for the insightful comments and hope that he/she reconsider the evaluation in light of our clarifications.
>
> The reviewer is correct in stating that one key difference of our approach with teacher-student (TS) training is that the current approach allows for weight sharing across the two networks. However, this is not the only difference. Another very important distinction is that in the current approach both the teacher and the student are trained together. While in the teacher-student setting the teacher is fixed and only the student learns. In the adjoint paradigm, the teacher also learns from the student. The advantage of this joint training is two fold. Table 1 shows that trained in this way, the smaller network performs better if it was trained in the TS fashion. Another very important advantage (as we see in Table 2) is that the bigger network also gains in performance when trained in the adjoint fashion. Hence, this way of training regularizes the bigger network as well, as is evident from Table 2 and Fig 3.
>
> Another point, which has also been raised by other reviewers, is comparison against other works. We have added a section in the current version which compares our approach against latest pruning/compression techniques.
>
> Method, Top-1 accuracy difference, Inference speed-up, Parameter reduction, Reference
>
> ABCPruner, 2.15, 2.18X, 2.17X, https://arxiv.org/pdf/2001.08565.pdf
>
> GBN, 0.67, 2.22X, 2.14X, https://arxiv.org/pdf/1909.08174.pdf
>
> DCP, 1.06, 2.25X, 2.06X, https://papers.nips.cc/paper/2018/file/55a7cf9c71f1c9c495413f934dd1a158-Paper.pdf
>
> HRank, 4.17, 2.63X, 1.86X, https://arxiv.org/pdf/2002.10179.pdf
>
> HRank, 7.05, 4.17X, 3.08X, https://arxiv.org/pdf/2002.10179.pdf
>
> Adjoint Network, 1.57, 3.1X, 13.7X, Current work
>
> The table above shows the difference in the top-1 accuracy between the standard network vs the network trained and pruned/compressed using the method referenced on the Imagenet dataset using the ResNet-50 architecture.
>
> We thank the reviewer for pointing out the reference "Rocket Launching: A Universal and Efﬁcient Framework for Training Well-Performing Light Net" by Zhou et al. We have added a reference to it the current draft. The boosternet architecture has two networks and the first few layers are shared across the same. While the adjoint network design also has weight sharing but we would like to point out that there are important differences in the way the weights are shared (Fig 1 and Fig 2). At each layer, some of the weights are shared while in the booster net, the entire layer is shared. We are also able to experimentally validate our approach on large datasets like Imagenet and on large architectures like Resnet50. Also, our loss function has some similarities to knowledge distillation loss but our loss function is also "time-dependent" and tunable $L = -y \log p + \lambda(t) KL(p, q)$. By choosing the  $\lambda$ function, we can choose to which rate we want the smaller network to learn from the larger network.

---

### Official Review · AnonReviewer1 · 2020-10-29
**An interesting approach to obtain a small CNN network while training a larger one.**

**Rating:** 5
**Confidence:** 4

**Review:**

The authors introduce the concept of "Adjoined Network" training, where a CNN network is trained concurrently to a much smaller network that is composed of a subset of the parameters of the larger network. The parameters are shared between the two, and the loss --aside from the standard cross-entropy of the larger network-- also incorporates the KL-divergence between the outputs of the two architectures. This is so the trained small network will be able to simulate the larger one with a fraction of the parameters, which makes for less memory and faster inference at deploy time.

The paper additionally shows that the concurrent training also acts as a regularizer for the larger network, and empirically demonstrates to be more effective than dropout.

In general, I like the general idea and how this adjoined training encourages the large network to be more careful at optimizing a specific subset of its weights. The improvement on the accuracy of the large network when trained adjoined is also a nice byproduct.

However, I have the following concerns:
- From a practical standpoint, this approach requires to train the large network alongside the small one. So, at least as described, this approach cannot be used with existing pre-trained networks which is the more typical scenario. It'd be interesting to see if a similar approach could be applied to obtain a small network by fine-tuning a pre-trained architecture.
- The results are marginally better than teacher-student, but the only architecture size being presented is the one for which the performance is comparable to that of the full network. I believe it's important to understand how the small adjoined network compares to the teacher-student network as a function of changing the desired size.

Minor: it would help the reader to add a paragraph describing intuitively which pieces of the larger CNN are shared by the smaller one.  From what I understand, in each layer, a small subset of the output channels is selected and additionally a random subset of such weights is not used?

---

> ### Author Response · Authors · 2020-11-25
> **Comparison against teacher-student**
>
> We thank the reviewer for the comments. The main concern of the reviewer seems to be how our proposed method compares to teacher-student network. Below, we address those concerns.
>
> ```The reviewer correctly points out that in the experiments presented, both the bigger and the smaller network are trained together from scratch. If we look at the loss function used for training.
>
> $L = - y \log p + \lambda(t) KL(p, q) $
>
> where $\lambda(t) = \min \{4t^2, 1\}$ where $t = 0$ at the start of the first epoch and $t = 1$ at the start of the last epoch during training. Hence, at the beginning only the larger network is trained and the training of the smaller network only kicks in after $t \ge 0.5$. Hence, using a pre-trained network corresponds to initializing the network differently and using a different regularization function $\lambda(t)$. We have added this experiment in the current draft and we see similar gains and the same trends as in Table 1. This suggests (as the reviewer rightly suspected) that the approach is equally applicable for pre-trained networks too.
>
> The second concern that the reviewer had was regarding the comparison against the smaller network trained in a teacher-student fashion. From Table 1, we see that the smaller network trained in the adjoint fashion consistently performs 1-2% better in terms of absolute accuracy as compared against the teacher-student mode of training. This gain is consistent across different datasets and two different architectures. We have also carried experiments across different sizes. We had not included this result in the current version due to space constraints. Below, is snapshot of the results comparing teacher-student with adjoint training on different compression sizes. The detailed results are included in the updated appendix.
>
> Network, Dataset, Mask Matrix (M), Adjoint small, Teacher student
>
> Resnet 50,  Cifar-100, a4, 64.73, 63.57
>
> Resnet 50,  Cifar-100, a8, 65.37, 63.57
>
> Resnet 50,  Cifar-100, a16, 65.11, 63.62
>
> Resnet 50,  Cifar-10, a4, 90.25, 88.93
>
> Resnet 50,  Cifar-10, a8, 89.77, 89.54
>
> Resnet 50,  Cifar-10, a16, 89.88, 89.08
>
> The main conclusion is that the network trained in an adjoint fashion performs better as compared against the same network trained in the teacher-student way even for different compression sizes.
>
> Regarding the parameters shared, if we look at Fig 2; at the beginning of training a small fraction ($\alpha$ fraction) of the convolution filters are selected and shared. This is represented by the mask matrix  $a_4$ (for example denoting that only 1/4th of the filters are selected and shared). In addition, a random subset is not used represented by the matrix $r_{0.8}$ (for example denoting that 80% of the weights are dropped).

---

### Official Review · AnonReviewer5 · 2020-11-08
**Promising work but some gaps**

**Rating:** 4
**Confidence:** 4

**Review:**

The authors present a very interesting idea of training a large network and a small network simultaneously with an interesting new loss function. The authors show that this can lead to a much smaller network with good accuracy (compared to the original network); thus this may be a good technique for sparsification.

**Theory**

Please clarify what induced L2 penalty means. The use of primes to indicate differentiation was ambiguous. Please be explicit about what you are differentiating with respect to in the notation.

Can you motivate why we should focus on the second order term during training?

Also if at all possible would be good to show the relevant quantities during training to see if the intuition is correct.

**Experimental Evidence**

Error bars from multiple runs to get a sense of variation relative to the difference being measured in Table 1 seems critical.

In addition, showing data from the sparsity literature on what other methods are able to accomplish (particularly weight magnitude pruning) would be very useful. For example, [Blalock and Gonzalez: What is the State of Neural Network Pruning?] seems to have data for ResNet-50, but it may not exactly apply, so the ideal thing would be to provide data by doing the baseline comparison yourself. (That paper also has a checklist of things to be careful of when doing pruning research.)

Also, a claim in the introduction is that training is faster this way since pruning and finetuning don’t need to be iterated. Would be good to back this up with actual numbers.

**Writeup**

The write-up could do with some more attention e.g. use \citep instead of \citet in most places. The mask M should be a tensor not a matrix. Also in your context, “speed” could refer to training time or inference latency. Would be nice to use these more canonical terms instead of just speed.

**Future Work**

One question that occurred to me while reading your paper (particularly the theory section) is what happens if you do the joint training as you describe, but instead of the adjoint loss, you use some sort of regular weight loss and different amounts of regularization penalties for the common weights and the weights unique to the large model. Would that give similar results?

For practical use of course, ResNet50 may not be the best target for sparsification since EfficientNets and MobileNets are already so much smaller. Please see the [Blalock and Gonzalez] paper (Sec 3.3) for perspective on this.

**Post Rebuttal**

Thanks for the response. I still didn't get an answer for why the second order term and not the first order term is of interest in the Taylor expansion. Also the paper does not seem to be updated with the promised changes, particularly error bars. So I am leaving my score as is.

---

> ### Author Response · Authors · 2020-11-24
> **Comparison against other works**
>
> We thank the reviewer for the insightful comments. We hope that the reviewer will reconsider the evaluation in light of our response below.
>
> $\textbf{Theory}$
>
> The taylor expansion of any function $f(x) = f(x_0) + \|x - x_0\|f'(x_0) + \frac{\|x-x_0\|^2}{2} f''(x_0) + O(\epsilon)$. The adjoint loss function $L = -y \log p + \lambda KL(p, q)$. The regularization behaviour is induced by the KL divergence term.  For any loss function, the L2 regularized loss function is $L = - y \log p + \lambda \|w\|^2$. Hence, by looking at the second order term in the taylor expansion of $f = KL(p, q)$, we can get the induced regularization behaviour.
>
> The use primes denotes differentiation. In our context, it refers to the partial derivative of the function (for example $\log p_i$) w.r.t to the given parameter $x$ or the parameter $y$. We thank the reviewer for pointing out this ambiguity. We have added a note in the paper addressing this issue.
>
>
> $\textbf{Experiments}$
>
> We did experiments over 3 runs and averaged the numbers. However, due to space constraints we did not add the error bars in the paper. Another reason why we did not add those error bars was that the results were fairly stable across multiple runs. We will add those bars the supplementary section.
>
> We agree with the reviewer that reporting numbers from other methods would be useful. While we do compare our method against other methods in the introduction and other sections, we did not add numbers in the experimental section. We concede that this was missing in the current version. We have added the following table in the current version of the paper which shows that our method significantly outperforms other methods for the architectures considered.
>
> Method, Top-1 accuracy difference, Inference speed-up,  Parameter reduction, Reference
>
> ABCPruner, 2.15, 2.18X, 2.17X, https://arxiv.org/pdf/2001.08565.pdf
>
> GBN, 0.67, 2.22X, 2.14X, https://arxiv.org/pdf/1909.08174.pdf
>
> DCP, 1.06, 2.25X, 2.06X, https://papers.nips.cc/paper/2018/file/55a7cf9c71f1c9c495413f934dd1a158-Paper.pdf
>
> HRank, 4.17, 2.63X, 1.86X, https://arxiv.org/pdf/2002.10179.pdf
>
> HRank, 7.05, 4.17X, 3.08X, https://arxiv.org/pdf/2002.10179.pdf
>
> Adjoint Network,	1.57, 3.1X, 13.7X, Current work
>
> The table above shows the difference in the top-1 accuracy between the standard network vs the network trained and pruned/compressed using the method referenced on the Imagenet dataset using the ResNet-50 architecture. We have added a experimental section in the main version of the paper.
>
>
> $\textbf{Formatting and future work}$
>
> We thank the reviewer for pointing out some of the formatting issues. We have fixed that in the current draft. We also thank the reviewer for suggesting some avenues for possible future work.

---

### Decision · Program_Chairs · 2021-01-07
**Final Decision**

**Decision:**

Reject

**Comment:**

The authors proposed to train a large network and a small network simultaneously with a new loss function. The parameters are shared between the two networks, and the loss also incorporates the KL-divergence between the outputs of the two models. In this way, the authors claim that one can train a small network with similar accuracy to the large one, while using less memory and having faster inference speed.

The reviewers think the papers is at the borderline. It has some interesting results, however, it also has quite a few problems:
1)	The technical novelty of the paper as compared to the teacher-student models is not adequate.
2)	There are many missing references and baselines, since model compression has been a long-studied problem.
3)	Experiments on more different NN models are preferred in order to verify the generalization ability of the proposed approach
4)	The compatibility with the pretraining framework is not very clear

The authors provided their rebuttals to the review comments. However, according the discussions among the reviewers, their concerns were not fully addressed yet and most of them would like to stand on their original scores. As a result, we do not think the paper should be accepted in its current form.